# Efficacy of a randomized controlled trial of a perinatal adaptation of COS-P in promoting maternal sensitivity and mental wellbeing among women with psychosocial vulnerabilities

Katrine Røhder[1][☉]*, Anne Kristine Aarestrup[2][☉], Mette Skovgaard Væver[3], Rikke Kart Jacobsen[2], Michaela L. Schiøtz[2]

1 Department of Psychology, University of Copenhagen, Copenhagen, Denmark, 2 Center for Clinical Research and Prevention, Frederiksberg Hospital, Copenhagen, Denmark, 3 Centre for Early Intervention and Family Studies, Department of Psychology, University of Copenhagen, Copenhagen, Denmark

☉ These authors contributed equally to this work.
* katrine.rohder@psy.ku.dk

**Data Availability Statement:** The participants in this study have not given consent to make these data publicly available. Therefore, the data and the

## Abstract

Pregnant women with psychosocial vulnerabilities should be offered perinatal interventions that include a parenting component to ameliorate the potential negative effects of maternal mental health problems and/or poor social network on parenting. One such intervention program is the Circle of Security–Parenting intervention (COS-P). The COS-P is a manualized video-based intervention that based on attachment theory seek to enhance maternal sensitivity and decrease the risk on insecure and disorganized attachment. We carried out a randomized controlled trial examining the efficacy of a perinatal adapted version of COS-P for women with psychosocial vulnerabilities (e.g. histories of mental health problems and/or poor social networks). Eligible participants ($N = 78$) were recruited to the study by midwives during regular prenatal sessions. Interventions were delivered individually at home by trained health nurses both pre and post birth. The primary outcome was maternal sensitivity assessed with the Coding Interactive Behavior Manual by blinded coders from video-recordings of mother-infant free play interactions. Secondary outcomes were mother-reported depressive symptoms, parental reflective functioning, parental stress, infant socio-emotional functioning, and maternal wellbeing. All outcomes were assessed at nine months infant age. We did not find an effect of the intervention on the primary outcome of maternal sensitivity ($\beta$ = -0.08; 95% CI [-0.41, 0.26], $p$ = .66). Neither did we find intervention effects on the secondary outcomes of depressive symptoms, parental reflective functioning, maternal well-being, or infant socio-emotional functioning. We did however find that the intervention decreased parental stress ($\beta$ = -8.51; 95% CI [-16.6;-0.41], $p$ = .04). The results are discussed in light of existing findings on the effect of COS-P and sample heterogeneity. Furthermore, we discuss the challenges of adapting the COS-P for pregnant women, some without prior experiences with caregiving. Future research with larger at-risk samples

information regarding the participants cannot be publicly available. A request for access to the data needs approval from appropriate Danish authorities and are subject to Danish regulations on personal data protection. A request for arrangement of data transfer agreements can be sent to the Center for Clinical Research and Prevention, Frederiksberg Hospital, Capital Region of Denmark, Copenhagen, Denmark (e-mail: tfe@regionh.dk) for researchers who meet the criteria for access to confidential data.

**Funding:** The entire project (design, data collection, data analysis, interpretation of data and interpretation of data) is funded by The Capital Region of Copenhagen. A representative from The Capital Region was part of the steering committee of the project. The steering committee support the implementation of the intervention. The steering committee was not involved in data collection, data analysis, interpretation of data, writing the report nor the decision to submit the report for publication.

**Competing interests:** The authors have declared that no competing interests exist.

examining moderation factors (e.g. adult attachment, depression maternal-fetal attachment) are recommended.

## Introduction

The foundation of children's physical, mental, and cognitive development is established before birth and affected by parental social and psychological risk factors [1]. Recent meta-analyses have documented that maternal wellbeing during pregnancy directly and indirectly affects the development of both fetus and child. As such, psychological stress and depression in pregnant women increase the risk of premature birth and low birth weight [2–4] and are indirectly related to later developmental problems in the child, such as altered stress response, attention deficit hyperactive disorder, depression, and anxiety, by affecting the mothers parenting skills [5–8]. Women with histories of mental health problems and/or poor social networks are at particular risk of relapse during the perinatal period with increased risk of depression and anxiety [9]. Finally, studies have shown that maternal wellbeing during pregnancy affects the pregnant woman´s emotional involvement with her unborn infant—a prerequisite for later parenting skills [10].

Early parenting behavior is considered the foundation for child development, with numerous studies demonstrating that "sensitivity" is one of the strongest predictors of positive developmental outcomes for children across different domains [11–14]. Sensitivity has been defined by Ainsworth and colleagues as a parent´s ability to 1) notice child signals, 2) interpret these signals correctly, and 3) respond promptly and appropriately to these signals [15]. Parental mental health problems are associated with an increased risk of lower sensitivity [16, 17] with sensitivity mediating the association between antenatal mental health problems and behavioral problems in the child [18, 19]. A related concept is parental reflective functioning which is defined as the parent's capacity to reflect upon her child's emotional experience and her own experiences as a parent [20]. Maternal reflective functioning capacities has been linked to caregiving behavior (e.g. sensitivity) and studies have demonstrated that higher levels of reflective functioning are associated with secure child attachment, better emotion regulation abilities, and less psychopathological symptoms [21]. Perinatal interventions for pregnant women with psychosocial vulnerabilities should include an interdisciplinary and intersectoral collaboration between general practice, the hospital, and the municipalities focusing on communication, coordination, and joint interventions to prevent gaps between pre- and postnatal services [22]. Furthermore, only treating psychopathology among mothers as an isolated intervention does not necessarily lead to improvement in the quality of parental behavior. This suggests that treatment of mental health problems should be accompanied by interventions that focus on supporting vulnerable parents in enhancing a good relationship with their child [23, 24]. Such early interventions should be initiated during pregnancy and continue into the postnatal period [25].

A recent systematic review and meta-analysis found that perinatal interventions for women with antenatal depression decrease offspring risk of psychopathology [26]. Core components of effective perinatal interventions for infants of mothers with mental health difficulties are: Facilitation of positive mother-infant interactions, helping mothers to understand the infant´s perspective, and where applicable the use of video-feedback on parent-infant interactions [27]. One such intervention is the Circle of Security-Parenting Intervention (COS-P). The COS-P is a manualized attachment-based video intervention designed to increase caregiver sensitivity to

child distress and ameliorate the risk of insecure and disorganized attachment especially among at-risk populations. The program was developed from the 20-session long Circle of Security program (COS; 17) to be applicable in clinical practice and in accordance with meta-analytic evidence suggesting better effect of shorter than longer parenting interventions [28]. A recent meta-analysis of the effect of the COS and COS-P intervention [29] found medium effect sizes for improving child attachment security (Hedge´s $g = 0.65$, $p = .003$), quality of caregiving (Hedge´s $g = 0.60$, $p = .012$), and reduction of caregiver depression (Hedge´s $g = 0.53$, $p < .001$), and a large effect size for improving caregiver self-efficacy (Hedge´s $g = 0.98$, $p < .001$). Despite its wide dissemination, there remains a need for studies on the effect of the COS-P [30, 31]. While the COS and the COS-P are usual carried out with pre-school children (mean age 34 months; 18), a perinatal version of COS has been examined among pregnant women with a history of substance use in jail for nonviolent offences [32]. This study found promising results with infant attachment status and maternal sensitivity resembling low risk groups. After the intervention, 70% of the study children were securely attached while 20% of them were classified as insecure-disorganized. There were no difference in maternal sensitivity between mothers participating in COS ($M = 8.55$, $sd = 2.42$) and mothers from a community sample ($M = 7.33$, $sd = 2.26$) in a 10-minutes free play interaction. The study did not include a control group which limits the conclusions that can be made from this study. Our study is the first to examine the efficacy of a perinatal adapted version of the COS-P.

## Study aims and hypothesis

As described in the study protocol [33], the overall aim was to test the efficacy of an early inter-disciplinary, intersectoral intervention, including a perinatal adaptation of the Circle of Security-Parenting Intervention developed in collaboration with the study authors, for at-risk pregnant women delivered by midwives and health nurses. The broad intervention consisted of four components: 1) detecting symptoms of mental illness and initiating treatment if indicated, 2) initiating a health nurse-delivered, individual, attachment-based parenting program (COS-P) adapted for initiation during pregnancy, 3) supporting breastfeeding decision and duration with education on techniques and strategies for successful breastfeeding, and 4) sharing knowledge and organizing family treatment pathways across sectors to address potential gaps between pre- and postnatal care for pregnant women and new mothers. In this study, we present results regarding the COS-P.

The hypothesis was that the COS-P intervention would increase the primary outcome of maternal sensitivity in interactions with the infant nine months after birth. Secondary outcomes and hypotheses were that the intervention would lead to better infant socio-emotional development, better maternal reflective functioning abilities, reduced maternal stress, reduced depressive symptoms, and improved maternal wellbeing.

## Methods

### Study design and setting

We conducted a randomized controlled trial at Herlev-Gentofte Hospital in Denmark and its four affiliated municipalities: Ballerup, Gentofte, Herlev and Rødovre. Study enrollment took place from June 2017 to November 2018. The trial was registered at ClinicalTrials.gov with trial number NCT03190707 June 16[th] 2017 and carried out in accordance with the published trial protocol [33].

## Participants

Pregnant women with psychosocial vulnerabilities were identified as potential participants by referring midwives from the hospital if they were identified as having psychological or social vulnerabilities by the general practitioner or the hospital. This is in line with official Danish Health Care recommendations [34]. The group consists of women with complex psychological/psychiatric problems and/or difficult social problems with a need for interdisciplinary perinatal care. In addition, participants should be residing in one of the four participating municipalities. Exclusion criteria included the inability to speak or understand Danish, age less than 18 years, a previous child placed in care outside the family during the intervention period, or a current registered or known maternal ICD-10 diagnosis of active eating disorder, severe depression, psychosis, schizophrenia, bipolar disorder, and severe obsessive compulsive disorder. Women with these diagnoses are offered treatment in the existing psychiatric system. All participants provided written informed consent. The study was reported to The Committee on Health Ethics of the Capital Region of Denmark (protocol number: 17006186) and registered at the Danish Data Protection Agency (registration no.: CSU-2017-003).

## Procedures

A researcher contacted eligible participants that consented to be contacted by phone, followed by a home visit to provide further information on the project, obtain informed consent, and enroll interested women and their families in the project. Data collection took place at three time points: at study inclusion at 3–5 months of pregnancy, when the infant is 8 weeks old, and after the intervention has ended when the child is 9 months old. Only data from the baseline assessment and after the intervention are reported in this study.

## Randomization

Eligible pregnant women who received a baseline data collection visit and provided informed consent were block randomized 1:1 to either usual care (the control group) or the intervention. Randomization was stratified within each municipality to ensure an equal distribution of participants in the intervention group across municipalities. A research assistant generated an allocation list and a person unaffiliated with the project prepared envelopes indicating randomization status from the list. After data were collected at the baseline visit, project personnel opened the envelopes.

## Usual care

Women randomized to the control group received usual care for pregnant women with psychosocial vulnerabilities. This included three consultations with their general practitioner, two ultrasound examinations, and four to seven consultations with midwives depending on parity [34]. Postnatal infant examinations were performed regularly by the health nurse in the infant's home and include measuring growth and evaluating the infant's physical and emotional development [34]. During the child's first year, the health visitor examined the infant at least twice within the first three weeks after birth, at two months, at four months (for first-time mothers), and again at eight months. Additional counseling home visits to families with psychosocial vulnerabilities after birth are generally provided by health visitors in all municipalities, with the number and content depending on families' specific needs. In our study, post hoc analysis revealed that mothers receiving usual care had had three more such visits from their health nurse compared to mothers in the intervention group suggesting a perceived need for extra interventions in our at-risk sample.

## Intervention

Women in the intervention group and their partners were in addition to usual care offered to participate in an early extended intervention (described above) including a perinatal adaptation of the COS-P intervention [35]. The adaptation consisted of an alteration of the order of when chapters were presented to the parents (see Table 1). The COS-P were delivered individually by their health nurse at two home visits before birth (using chapter 1 and chapter 3 from COS-P) and seven home visits after birth (using the COS-P chapters in the following order: chapter 4; chapter 2; chapter 3; chapter 5; chapter 6; chapter 7 and chapter 8). Health nurses were certified in the use of COS-P during a four-day intensive training course, and a COS-P supervisor ensured adherence to the COS-P manual.

## Outcomes

The primary outcome was maternal sensitivity, assessed during a five-minute mother-infant interaction sequence coded with the Coding Interactive Behavior Manual (CIB) [36]. The CIB is a global rating scheme for coding interactions between parent and infants aged 2 to 36 months. The CIB at nine months consists of 33 items that are aggregated into composites of maternal sensitivity and intrusiveness, infant involvement and withdrawal, dyadic reciprocity, and dyadic negative states. Each items is coded on a 5-point scale with high scores representing many observations of the given behavior or affective state being rated. There are no norms for the CIB. The CIB has been validated in both longitudinal and intervention studies showing predictive validity for child development outcomes [37, 38], discriminate validity between mothers with and without mental health problems [17], as well as ability to detect effects of perinatal interventions [39, 40]. The current study used the maternal sensitivity composite. At approximately nine months, mother-infant interactions were videotaped during a home visit by a researcher not blinded to mothers' allocation to the experimental vs. control group. A coder trained in the CIB by Ruth Feldman blinded to treatment allocation coded all mother-infant interactions. Reliability was assessed with a randomly selected subset of 23% of interactions coded by a second, allocation-blinded coder trained by Ruth Feldman. ICC estimates and their 95% confidence intervals were calculated using SPSS, version 25 (SPPS inc, Chicago, IL) based on single-rating, consistency, two-way random-effects model. Only items contributing to the overall composite, defined as item-total correlations of .3 or higher, were included in the study composite analyses [41].

Secondary outcomes collected from self-report questionnaires included the following measures.

*Infant' socio-emotional development* was measured with the Ages and Stages Questionnaire–Social Emotional (ASQ-SE) screening tool, which measures self-regulation, compliance,

**Table 1. Overview of the COS-P visits.**

| Visit | Time | COS-P module |
|---|---|---|
| 1 | Pregnancy week 32 | Module 1 –Introduction to the Circle |
| 2 | Pregnancy week 34 | Module 3 (first half of the module)–Managing your child's emotions |
| 3 | Baby at 9 weeks | Module 4 –Organizing your child's feelings |
| 4 | Baby at 11 weeks | Module 2 –Exploring your children's needs |
| 5 | Baby at 13 weeks | Module 3 (second half of the module)—Managing your child's emotions |
| 6 | Baby at 22 weeks | Module 5 –The path to security |
| 7 | Baby at 24 weeks | Module 6 –Exploring our struggles |
| 8 | Baby at 26 weeks | Module 7 –Rupture and repair in our relationships |
| 9 | Baby at 36 weeks | Module 8 –Summary and celebration |

communication, adaptive functioning, autonomy, emotions, and interaction with other people [42]. It consists of 22 questions regarding the infant´s behavior and the parent is asked to indicate if that behavior happens often or always, sometimes, or rarely or never. In addition, the parent can indicate if the behavior is of concern of the parent. Answers are scored and summed afterwards on a scale from 0 to 45 with high scores representing more problematic socio-emotional behavior. The ASQ-SE is a comprehensive and psychometrically sound measure of infant socio-emotional development [43].

*Parental reflective functioning (PRF)* is the capacity to focus on one's own and the child's experience and feelings. It was measured with three subscales of the Parental Reflecting Functioning Questionnaire (PRFQ) [44]: 1) interest and curiosity in mental states (Interest and curiosity; IC), 2) ability to recognize the opacity of mental states (certainty of mental states; CM), and 3) non-mentalizing modes characteristic of parents with impaired PRF (prementalizing; PM) (e.g, malevolent attributions, inability to enter the subjective world of the child). There are no total score. Each subscale consist of 6 statements that are rated on a 7-point likert scale from "complete agree" to "complete disagree". High scores on factor 1 and 2 represent high levels of parental reflective functioning although very high scores might be associated with hyper-mentalization. Low scores on factor 3 are considered optimal. Validation studies of the PRFQ provide evidence for its reliability and validity [45, 46]. The version used in this study was translated by Mette Skovgaard Væver and Johanne Smith-Nielsen, Center for Early Intervention and Family Studies, University of Copenhagen.

*Parenting stress* was measured with the 36-items short form of the Parenting Stress Index (PSI-SF) comprising three subscales—parental distress, parent-child dysfunctional interaction, and difficult child—in addition to the total stress scale [47]. The total scale ranges 36 to 180, while the subscales range from 12 to 60. High scores represent high stress. Studies provide psychometric support for the PSI-SF as an effective and appropriate measure for use with high-risk families [48].

*Maternal depressive symptoms* were assessed with the Edinburgh Postnatal Depression Scale (EPDS). The EPDS is a self-report questionnaire that is widely used to detect postnatal depression. It consists of 10 items assessing depressive symptoms during the past week (range 0–30). High scores indicate high depressive symptoms. EPDS has high sensitivity and specificity for detecting depression, based on clinical psychiatric diagnostic criteria [49, 50].

*Parental wellbeing* was measured using the Short Warwick Edinburg Mental Well-being Scale (SWEMWBS), which reflects a broad understanding of well-being that includes affective-emotional aspects, cognitive-evaluative dimensions, and psychological functioning. The SWEMWBS consists of 7 items scored from 1 ("none of the time") to 5 ("all of the time"). Afterward scores are converted to metric scores using on online conversion table. All items are worded positively and address aspects of positive mental health [51, 52]. We used the Danish translation by Line Nielsen, Carsten Hinrichsen, Ziggi Ivan Santini and Vibeke Koushede from University of Southern Denmark.

## Statistical analysis

Power analysis was conducted prior to recruitment and described in our protocol paper [33]. The CIB uses an interval scale from 1 to 5. We assumed that the average mean of CIB-maternal sensitivity would be 3 with a standard deviation of 0.9. One study found a difference of 0.9 between mothers with and without depression. We expected a between-group difference of 0.75. Based on a t-test, 80% power and a two-sided significance level $\alpha$ of 0.05 and therefore we aimed to include 48 families with 24 per group. We estimated a likely dropout rate of 20% and therefore aimed to include 60 families (30 in each group) at randomization.

Descriptive data were analyzed as means with standard deviations, medians with interquartile ranges, or frequencies with percentages, depending on the distribution of variables.

An ANOVA analysis of the primary outcome of maternal sensitivity in the intervention and control groups was performed, with the effect measured as the between-group difference at nine months of infant age. The unadjusted primary analysis followed the intention-to-treat principle, using multiple imputation for missing values. Secondary, the analysis was adjusted for background variables (parity, education, maternal age, baseline parental measures) due to prior hypothesis regarding their impact on the outcome and analyzed using the per protocol principle; only participants with high compliance rates (participation in seven out of nine Circle of Security sessions) were included in the intervention group. To assess whether the intervention effect was diluted by including non-completers (intervention group participants participating in four or fewer COS-P sessions), the analyses described above were repeated with non-completers excluded from the intervention group. Both multiple imputation and regression analyses were repeated with municipality included due to the stratified design of the study. Municipality was included/modelled as a random effect in mixed models. Similar analyses were performed on secondary outcomes. For secondary analysis of variables with measured baseline values, an ANCOVA model of changes in outcome from baseline to follow-up adjusted for baseline values was used.

## Results

A total of 158 women were invited to participate in the study (Fig 1); of these, 61 chose not to participate and 19 did not meet inclusion criteria. The most frequent reasons for not wanting to participate in the study were "do not experience a need for extended services", "cannot manage to participate due to vulnerabilities or due to the extra time it will require", and "do not want to be filmed". Women who did not meet inclusion criteria had moved or planned to move out of the municipality (8), miscarried (6), did not speak and understand Danish (2), were younger than 18 years (1), wanted to give birth at another hospital (1), or lacked the mental capacity needed to indicate consent (1). All women who accepted a visit from one of the researchers, a total of 78 women and their families, consented to participate in the study and were randomized. Two participants had subsequent miscarriages and were omitted from analyses. In addition, 62 fathers/partners participated in the study, although their data are not included in this report.

### Participant characteristics

More than half of study participants were pregnant with their first child, nearly all were married or had a partner, and 86% had at least some post-primary education (see Table 2). Almost half of participating women had a mental health problem when they became pregnant, which more than a third received psychiatric or psychological treatment for. Almost 90% had previously received psychiatric or psychological treatment. More than 85% of the participants reported having a strong social network from which they often or always experienced support. However, a greater proportion of participants in the intervention group were employed or students, while more participants in the control group were on sick leave or received social support.

### Intervention effects

The primary outcome was maternal sensitivity measured with the CIB. Internal consistency of the original CIB-sensitivity composite was calculated (Cronbach's alpha $\alpha$ = .87). Reliability for the maternal sensitivity composite (acknowledging, imitating, elaborating, positive affect,

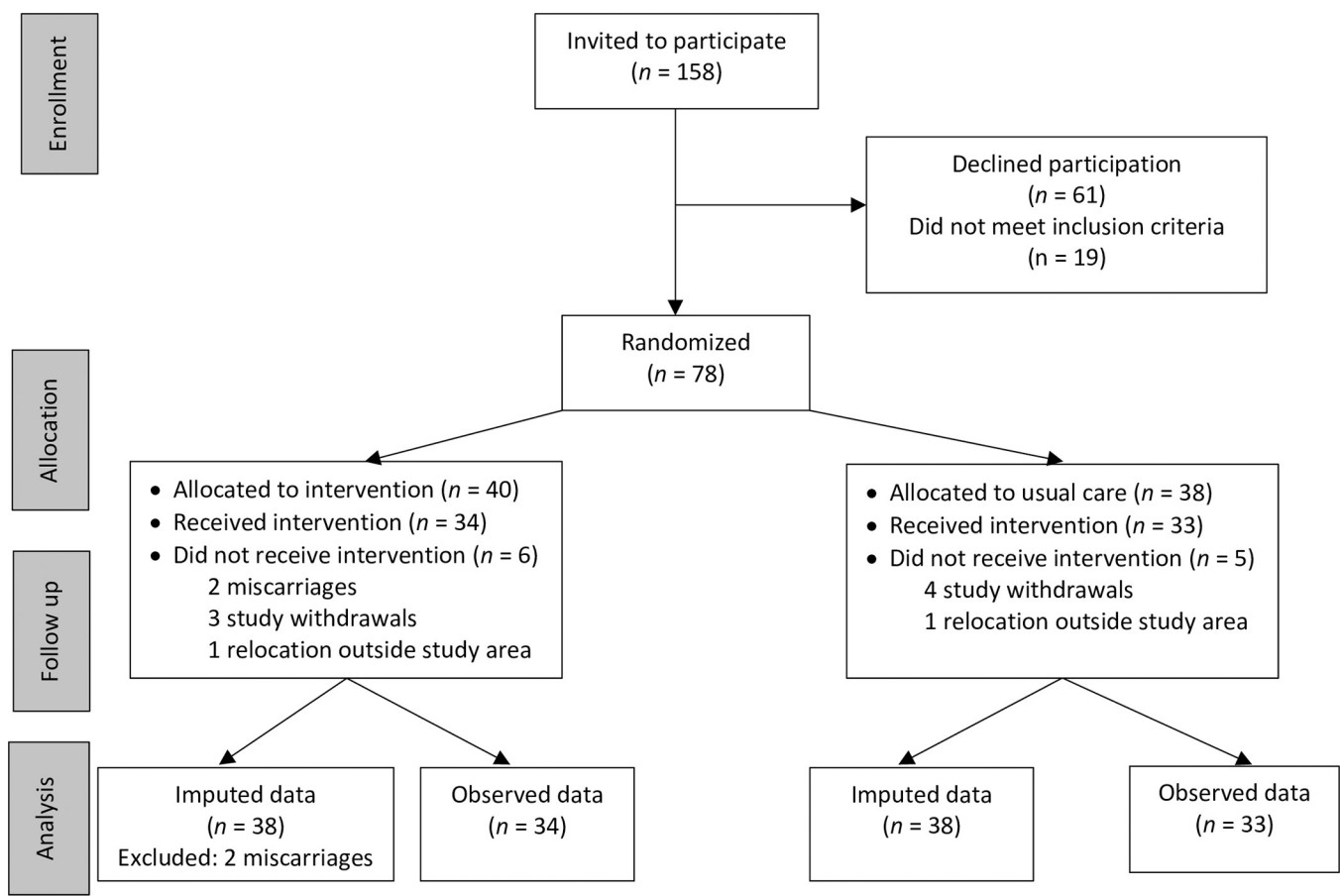

**Fig 1. Flowchart of participants.**

**Table 2. Participant characteristics.**

|  | All women $N = 76$ | Control $n = 38$ | Intervention $n = 38$ |
|---|---|---|---|
|  | n (%) | n (%) | n (%) |
| Pregnant with first child | 40 (52.6) | 19 (50.0) | 21 (55.3) |
| Married/partner | 65 (85.5) | 32 (84.2) | 33 (86.8) |
| Primary | 11 (14.5) | 5 (13.2) | 6 (15.8) |
| Vocational or short secondary (1–2 years) | 30 (39.5) | 14 (36.8) | 16 (42.1) |
| Medium length or long secondary (3–7 years) | 35 (46.1) | 19 (50.0) | 16 (42.1) |
| Employed | 51 (67.1) | 23 (60.5) | 28 (73.7) |
| Student | 7 (9.2) | 1 (2.6) | 6 (15.8) |
| On sick leave/ social support | 18 (23.7) | 14 (36.8) | 4 (10.5) |
| Mental health problems at pregnancy | 35 (46.1) | 17 (44.7) | 18 (47.4) |
| Current psychiatric or psychological treatment | 28 (36.8) | 14 (36.8) | 14 (36.8) |
| Previous psychiatric or psychological treatment | 67 (88.1) | 35 (92.1) | 32 (84.2) |
| Strong social network (often or always support from partner and family or friends) | 65 (85.5) | 34 (89.5) | 31 (81.6) |
|  | M (SD) | M (SD) | M (SD) |
| EPDS | 8.50 (5.4) | 8.50 (5.14) | 8.34 (5.13) |

**Table 3. Regression coefficients from ANOVA and ANCOVA for association of the COS-P intervention on primary and secondary outcomes at infant age nine months.**

| | Control *n* = 33 | Intervention *n* = 34 | Between-group difference | |
|---|---|---|---|---|
| | *M (se)* | *M (se)* | *β*, 95% CI | *p* |
| Maternal sensitivity (CIB) | 3.5 (0.8) | 3.4 (0.6) | -0.08 (-0.41, 0.26) | .66 |
| Child socio-emotional development (ASQ-SE) | 39.8 (18.1) | 34.8 (20.2) | -4.91 (-14.16, 4.35) | .30 |
| Parental reflective functioning (PRFQ) | | | | |
| Interest and curiosity in mental states | 6.1 (0.8) | 6.0 (0.7) | -0.13 (-0.50, 0.24) | .49 |
| Pre-mentalizing | 1.6 (0.6) | 1.4 (0.4) | -0.21 (-0.44, 0.02) | .07 |
| Certainty of mental states | 4.4 (1.1) | 4.4 (1.0) | 0 (-0.52, 0.51) | .99 |
| Parental stress (PSI-SF) | | | | |
| Total | 68.5 (19.5) | 59.9 (14.2) | -8.51 (-16.6, -0.41) | .04 |
| Parental distress | 26.5 (8.7) | 23.2 (6.5) | -3.31 (-6.95, 0.33) | |
| Parent-child dysfunctional interaction | 19.9 (8.3) | 16.6 (4.5) | -2.94 (-5.98, 0.09) | |
| Difficult child | 22.3 (7.1) | 20.2 (5.4) | -1.97 (-4.83, 0.88) | |
| Depressive symptoms (EPDS)* | -2.0 (4.5) | -2.7 (5.1) | -0.71 (-2.35, 0.91) | .39 |
| Well-being (SWEMWBS)* | 1.4 (3.6) | 1.1 (2.8) | -0.11 (-1.45, 1.23) | .87 |

*Note.* Data presented from imputed data.

* ANCOVA model of changes in outcome from baseline to follow-up adjusted for baseline values Abbreviations: ASQ-SE, Ages and Stages Questionnaire-Social Emotional; CIB, Coding Interactive Behavior; EPDS, Edinburgh Postnatal Depression Scale; PRFQ, Parental Reflecting Functioning Questionnaire; PSI, Parental Stress Index; SWEMWBS, Short Warwick Edinburgh Mental Well-Being Scale.

vocal appropriateness, appropriate range of affect, resourcefulness, praising, parent supportive presence) was good; ICC = .749 [.667;.813].

Table 3 displays means and standard errors for the primary and secondary outcomes at nine months in the intention-to-treat analysis with imputed data. The results did not change when including only observed data or in per-protocol analyses. No statistically significant between-group differences were observed for maternal sensitivity, child socio-emotional development, parental reflective functioning, maternal depressive symptoms, or parental well-being. However, mothers in the intervention group experienced lower parental stress to a statistically significant degree.

## Discussion

The study aim was to examine the efficacy of an early interdisciplinary, intersectoral intervention, including the parenting program COS-P, for at-risk pregnant women. No statistically significant differences were found between groups for the primary outcome measure of maternal sensitivity or the secondary outcome measures of parental reflective functioning, depressive symptoms, well-being, and duration of breastfeeding. Mothers in the intervention group reported lower levels of parental stress at nine months of infant age, compared to mothers in the control group.

### Maternal sensitivity

The results did not support our hypothesis that the intervention would increase maternal sensitivity. Although some studies have found a positive effect of COS-P on observed maternal emotional availability and parent-reported parenting behavior [53, 54], others have not been able to replicate this finding [55, 56]. We assessed maternal sensitivity using a five-minute video recording of an unstructured mother-child interaction. One potential explanation for

the lack of difference between intervention and control groups is that we assessed maternal sensitivity in a stress-free environment. Cassidy et al. [56] found a main effect of COS-P on maternal response to child distress indicated by reduced unsupportive responses. Thus, the lack of any detected effect on maternal sensitivity could be explained by the fact that most of the mothers were able to be sensitive over a short period in a situation without specific require-ments, which is consistent with other studies [57, 58].

Another explanation for the lack of an intervention effect on maternal sensitivity may be that mothers in the control group received care as usual. This included extra care and support provided by municipalities, which may also focus on supporting better mother-infant interac-tions. Indeed, mothers receiving usual care received three more visits from their health nurse than did mothers in the intervention group.

A third potential explanation for the lack of a main effect on maternal sensitivity is that mothers benefit differently from the Circle of Security program. Cassidy et al. [56] indicated that COS-P may be particularly beneficial for mothers who are high on attachment avoidance or low on depressive symptoms, and Maxwell et al. [30] found within-treatment group effects showing greater reduction in caregiving helplessness for mothers with older children and a greater reduction in hostility for mothers with pre-intervention clinical levels of depression. Ramsauer et al. [55] found that maternal unresolved state of mind regarding attachment mod-erated changes in maternal sensitivity in the intervention group, as compared to treatment as usual. It would have been interesting to explore 'what works for whom' in this study, but the relatively low sample size yielded insufficient power to identify potential moderating effects of the intervention.

Finally, means and standard deviations from both groups indicate considerable individual variation among participants. Some mothers had a high degree of sensitivity and were sensitive in interactions with their child for the greater part of the five-minute interaction, whereas oth-ers showed a low degree of sensitivity and missed many opportunities to respond appropriately to their child's signals. This indicates that the mothers in our sample, despite shared psychoso-cial vulnerabilities, are a heterogenous group in terms of parenting and that interventions focusing on improving parental sensitivity may not be needed by all such mothers. In a study of the prenatal risk status of the study sample [59], we found that, as compared to women from a community sample, "only" approximately half of the women demonstrated lower and subop-timal maternal-fetal bonding indicating prenatal and pre-intervention difficulties related to parenting.

## Parenting stress and reflective functioning

The intervention affected perceived level of stress related to parenting. Prior studies show that a high level of stress among parents impacts their ability to regulate their own emotions and, consequently, their parenting skills and capability to help the child regulate his/her emotions [60]. A lower level of reported parental stress in the intervention group may mean that the mothers' parenting skills improved and that they are more capable of helping their child with emotional regulation, which will lead in the long run to better emotional skills for the child. We found no intervention effect on parental reflective functioning, although the effect approached statistical significance for the pre-mentalizing mode. This subscale measures a non-mentalizing stance where the parent have difficulties entering the subjective world of the infant and may attribute malevolent intentions to the infant. We found that mothers in the intervention group had lower scores than mothers in the control group ($p = .07$).

Several studies have found that COS-P positively affects parental self-efficacy, defined as the parent´s sense of their own parenting abilities, which is comparable to a lower experience of

parenting stress [30, 53]. It has been argued that the COS-P intervention aims to enhance parents´ understanding of their child´s behavior and to develop their abilities to observe and reflect as well as help to understand their own thoughts and feelings in relation to parenting– that is, the *representational* level of parenting [30]. This might explain why we detected intervention effects on mothers' experiences of their parenting but not on observed behaviors.

### Depressive symptoms

We found no intervention effect on maternal depressive symptoms. As a group, mothers participating in the study reported subthreshold levels of depression. Maxwell et al. [30] found that, in the intervention group, mothers with probable clinical depression at baseline (defined in their study as an EPDS score $\geq$ 13) had a significantly greater reduction in depression scores than did mothers with baseline depression scores in the lower range.

### Infant socio-emotional development

We did not find a statistically significant between-group difference in mother-reported socio-emotional development of the child. Cassidy et al. [56] also found no main effects of the intervention on child attachment or behavioral problems but found that the effect was moderated by maternal attachment style or depressive symptoms.

Despite the at-risk status of all mothers in our sample, their infants´ socio-emotional development appears more heterogeneous, as reflected in ASQ-SE scores. Scores in both groups are just below subthreshold ($<$ 40), with standard deviations well into the clinical range. Some infants in the sample may have socio-emotional difficulties, while others do not. Future studies should explore risk and resilience factors for infant socio-emotional difficulties and what role the COS-P may have in ameliorating maternal risks factors for infant development.

### Limitations

The main limitation of the study is the small sample size precluding moderation analysis. Recent evidence suggest that COS-P works best for women with insecure attachment (attachment avoidance or unresolved state of mind). In addition, depressive symptoms seem to moderate intervention effectiveness although studies disagree on the level of depression (low levels vs. clinical levels). More research on the effect of the COS intervention is needed with large samples utilizing a 'what works for whom' approach.

Second, due to ethical reasons we were not able to control the content of usual care. It may be that women in the control group received parenting support by their health nurse that to some degree compensated for the intervention.

Finally, we applied the COS-P intervention in a perinatal setting. Despite adaptation to fit the pregnancy period, it may be that COS-P is more effective among parents of older children as the program focuses on enhancing parental perception and understanding of differences between the child´s attachment and exploration systems. Differences in these behavioral systems may not be manifest among newborns. Indeed, most studies have examined the intervention among parents of children older than 6 months.

### Conclusion

We carried out a RCT to examine the efficacy of an early interdisciplinary, intersectoral intervention for pregnant women with complex psychosocial vulnerabilities, including a perinatal adaptation of the COS-P intervention. The intervention reduced self-reported maternal stress in relation to parenting which may lead to improved parenting behavior. However, we did not

find an effect of the intervention of maternal sensitivity, infant socio-emotional development, parental reflective functioning, or depressive symptoms. More research on the effect of COS-P is needed with larger samples involving moderation analyses allowing conclusion to be made on 'what works for whom'.

## Supporting information

**S1 Checklist.**
(DOC)

**S1 File. Perinatal COS-P protocol registration clinical trials.**
(PDF)

## Acknowledgments

This study could not have been conducted without the participation and involvement the midwives and managing midwives from Herlev Hospital and the health nurses and managers from the municipalities of Ballerup, Herlev, Gentofte and Rødovre. For this we are very grateful. Also, we would like to thank all the participating families for their engagement and contribution to the study.

## Author Contributions

**Conceptualization:** Anne Kristine Aarestrup, Mette Skovgaard Væver, Michaela L. Schiøtz.

**Formal analysis:** Katrine Røhder, Rikke Kart Jacobsen.

**Investigation:** Anne Kristine Aarestrup, Michaela L. Schiøtz.

**Project administration:** Anne Kristine Aarestrup, Michaela L. Schiøtz.

**Writing – original draft:** Anne Kristine Aarestrup.

**Writing – review & editing:** Katrine Røhder, Mette Skovgaard Væver, Rikke Kart Jacobsen, Michaela L. Schiøtz.

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
