## [Decision Letter · Decision Letter 0]

7 Jan 2022

PONE-D-21-31294Efficacy of a Randomized Controlled Trial of a Perinatal Adaptation of COS-P in Promoting Maternal Sensitivity and Mental Wellbeing among Women with Psychosocial VulnerabilitiesPLOS ONE

Dear Dr. Røhder,

Thank you for submitting your manuscript to PLOS ONE. After careful consideration, we feel that it has merit but does not fully meet PLOS ONE’s publication criteria as it currently stands. Therefore, we invite you to submit a revised version of the manuscript that addresses the points raised during the review process.

We look forward to receiving your revised manuscript.

Kind regards,

Anna Palatnik, M.D.

Academic Editor

PLOS ONE

Journal Requirements:

2. Thank you for stating the following in the Acknowledgments/ Funding Section of your manuscript: 

The entire project (design, data collection, data analysis, interpretation of data and interpretation of data) is funded by The Capital Region of Copenhagen. A representative from The Capital Region was part of the steering committee of the project. The steering committee support the implementation of the intervention. The steering committee was not involved in data collection, data analysis, interpretation of data, writing the report nor the decision to submit the report for publication.

The entire project (design, data collection, data analysis, interpretation of data and interpretation of data) is funded by The Capital Region of Copenhagen. A representative from The Capital Region was part of the steering committee of the project. The steering committee support the implementation of the intervention. The steering committee was not involved in data collection, data analysis, interpretation of data, writing the report nor the decision to submit the report for publication.

Reviewers' comments:

Reviewer's Responses to Questions

**Comments to the Author**

1. Is the manuscript technically sound, and do the data support the conclusions?

Reviewer #1: Partly

2. Has the statistical analysis been performed appropriately and rigorously? 

Reviewer #1: No

3. Have the authors made all data underlying the findings in their manuscript fully available?

Reviewer #1: No

4. Is the manuscript presented in an intelligible fashion and written in standard English?

Reviewer #1: Yes

5. Review Comments to the Author

Reviewer #1: Abstract

- typo in line 20: 'woman' instead of 'women'.

- typo in line 28: 'midwifes' instead of 'midwives' - and in the rest of the manuscript, e.g. line 107.

- the results paragraph of the abstract should include some numerical results, specifically the estimates of effect for key outcomes with 95% confidence intervals and p-values; perhaps including a sentence on the total number of women recruited.

Background

- line 74: please report the magnitudes of the 'medium effect sizes' mentioned here.

- line 80: please state objectively what the 'promising results' were.

Methods

- Procedures - should be consistently reported in the past tense like the rest of the manuscript. The beginning is reported in past tense, but sections such as line 130-135 are in the present tense.

- Intervention - the 'perinatal adaptions' to the COS-P intervention mentioned in line 141 should be clearly described.

- the CIB outcome should be described in a more detail. It is not clear how it is scored (if it is in fact a score), what is the measurement scale, what are the normal ranges, does it have any components/items/domains and if so how are they scored, etc. The same detail should be included for the secondary outcomes.

- statements in lines 152-153 about the Cronbach's alpha and in 158 about the ICC are results and should be reported in the results section.

- the sample size is not well described, partly because the primary outcome is not well described either. If the primary outcome is continuous, then the sample size calculation should include some idea about what the expected mean and SD of the outcome in the control group (plus the difference to be detected). You should also state whether the level of significance is for a one-sided or two-sided test.

- as this is a randomised trial, you should not conduct tests comparing descriptive characteristics between the groups (line 197-199).

- for the same reason above, any adjustment in the analysis should not normally be based on whether there are differences in baseline characteristics; however, design factors such as the stratification variables should be adjusted for.

Results

- given my comments above regarding analysis of a randomised trial, the results on line 233-234 about between group differences, and the p-value column on Table 1 (and the footnote) should be removed.

- Table 1 should report means and standard deviations (SD) for continuous characteristics. It is unclear whether the quantities reported against EPDS are the mean and standard deviation. Table 1 should also be clear when reporting counts and proportions of categorical variables, and explicitly state this.

- Table 2 should report means and standard errors (SE) for continuous outcomes. The 'range' column is not necessary here, and should have been included in the description of the measures (see comments above)

- again in Table 2, for outcomes with domains/components, p-values should generally be reported for the total or overall score, not the components, as this constitutes multiple testing which then warrants corrections that are difficult do to in this small study.

- for outcome measures where you have baseline values, instead of conducting an analysis on the change-from-baseline measure, you should ideally conduct an analysis on the post-intervention measures and adjust for the baseline measures (i.e. analysis of covariance ANCOVA). This is the most statistically-efficient way to analyse these outcome.

6. PLOS authors have the option to publish the peer review history of their article (what does this mean?). If published, this will include your full peer review and any attached files.

Reviewer #1: No

---

## [Author Response · Author response to Decision Letter 0]

8 Mar 2022

Dear Dr. Palatnik and reviewer 1

Thank you for your comments and for giving us the opportunity to re-submit our manuscript.

We hope the changes made is sufficient and that you will consider accepting the manuscript for publication.

Katrine Røhder

---

## [Decision Letter · Decision Letter 1]

19 Aug 2022

PONE-D-21-31294R1

Efficacy of a Randomized Controlled Trial of a Perinatal Adaptation of COS-P in Promoting Maternal Sensitivity and Mental Wellbeing among Women with Psychosocial Vulnerabilities

PLOS ONE

Dear Dr. Røhder,

Thank you for submitting your manuscript to PLOS ONE. After careful consideration, we feel that it has merit but does not fully meet PLOS ONE’s publication criteria as it currently stands. Therefore, we invite you to submit a revised version of the manuscript that addresses the points raised during the review process.

As we only had comments from one reviewer in the first round of reviews, we sought additional input from a subject expert. The first reviewer is satisfied with your revisions, but the second reviewer has  some requests for clarification on a number of issues. Please see those comments below.

We look forward to receiving your revised manuscript.

Kind regards,

Steve Zimmerman, PhD

Associate Editor, PLOS ONE

Reviewers' comments:

Reviewer's Responses to Questions

**Comments to the Author**

1. If the authors have adequately addressed your comments raised in a previous round of review and you feel that this manuscript is now acceptable for publication, you may indicate that here to bypass the “Comments to the Author” section, enter your conflict of interest statement in the “Confidential to Editor” section, and submit your "Accept" recommendation.

Reviewer #1: All comments have been addressed

Reviewer #2: (No Response)

2. Is the manuscript technically sound, and do the data support the conclusions?

Reviewer #1: Yes

Reviewer #2: Partly

3. Has the statistical analysis been performed appropriately and rigorously? 

Reviewer #1: Yes

Reviewer #2: Yes

4. Have the authors made all data underlying the findings in their manuscript fully available?

Reviewer #1: No

Reviewer #2: Yes

5. Is the manuscript presented in an intelligible fashion and written in standard English?

Reviewer #1: Yes

Reviewer #2: No

6. Review Comments to the Author

Reviewer #1: In line 274 you have stated that the means and standard deviations are reported in Table 2. However the column titles of Table 2 appear to indicate that you are reporting standard errors (which is the correct thing to report here, as this is a table of inferential results). So either (1) correct the statement on line 274 or (2) report standard errors in Table 2 if they are in fact standard deviations and correct the statement on line 274.

Reviewer #2: The paper is a randomized controlled trial exploring the efficacy of the COS-P intervention, adapted to the perinatal period, in improving several aspects of mothers’ mental health and well-being. Despite the topic is interesting, there are several aspects that should be clarified, before resubmitting the paper:

1. The primary aim of the study is to increase of “maternal sensitivity in interactions with the infant nine months after birth”. The concept of “maternal sensitivity” is not really clear to me. No explanation of this is provided in the introduction and/or in the method section. Could the authors better specify what they intend for “maternal sensitivity”. Similarly, among secondary outcome it is reported that the intervention could lead to a “better maternal ability to mentalize”; could this issue be better described?

2. One of the inclusion criteria is that mothers had to present “psychosocial or social vulnerabilities” identified by the general practitioner. This is too vague and undefined. Please specify in detail the inclusion criteria, since this is a RCT and inclusion criteria are crucial to select the right population to be treated.

3. The experimental intervention is poorly described. Authors should explain in details the experimental intervention (i.e., how many sessions were provided to mothers, what was the content of sessions, how session were structured, etc.), and report the appropriate citation to the available manuals.

4. The main outcome was assessed through the CIB. Is this global rating scheme validated? Were raters trained in the use of the assessment instruments? Moreover, authors should specify whether mental health professionals who performed the assessments were blinded to mothers’ allocation to the experimental vs. control group.

5. I was not able to find thought the manuscript when women were assessed for the first time (I suppose during pregnancy) and when they were reassessed. Could the author better specify this issue?

6. In the result section, it is reported that 35 out of 76 women presented mental health problems at pregnancy and 28 of them were on psychiatric or psychological treatment. This is a bit confusing, since authors have excluded from the recruitment women with major psychiatric problems. Could you clarify this issue?

7. PLOS authors have the option to publish the peer review history of their article (what does this mean?). If published, this will include your full peer review and any attached files.

Reviewer #1: No

Reviewer #2: No

---

## [Author Response · Author response to Decision Letter 1]

26 Aug 2022

1. Reviewer #1: In line 274 you have stated that the means and standard deviations are reported in Table 2. However the column titles of Table 2 appear to indicate that you are reporting standard errors (which is the correct thing to report here, as this is a table of inferential results). So either (1) correct the statement on line 274 or (2) report standard errors in Table 2 if they are in fact standard deviations and correct the statement on line 274.

Thank you for spotting this. We are reporting standard errors in Table 2. We have corrected the statement on line 327.

Reviewer #2: The paper is a randomized controlled trial exploring the efficacy of the COS-P intervention, adapted to the perinatal period, in improving several aspects of mothers’ mental health and well-being. Despite the topic is interesting, there are several aspects that should be clarified, before resubmitting the paper:

1. The primary aim of the study is to increase of “maternal sensitivity in interactions with the infant nine months after birth”. The concept of “maternal sensitivity” is not really clear to me. No explanation of this is provided in the introduction and/or in the method section. Could the authors better specify what they intend for “maternal sensitivity”. Similarly, among secondary outcome it is reported that the intervention could lead to a “better maternal ability to mentalize”; could this issue be better described?

We added a section in the introduction on maternal sensitivity and parental reflective functioning (lines 60-71). Furthermore, we decided to use the wording “parental reflective functioning” instead of “mentalizing abilities” throughout the manuscript as this is more consistent and in line with the measure used (the P-PRFQ).

“Early parenting behavior is considered the foundation for child development, with numerous studies demonstrating that “sensitivity” is one of the strongest predictors of positive developmental outcomes for children across different domains (11-14). Sensitivity has been defined by Ainsworth and colleagues as a parent´s ability to 1) notice child signals, 2) interpret these signals correctly, and 3) respond promptly and appropriately to these signals (15). Parental mental health problems are associated with an increased risk of lower sensitivity (16, 17) with sensitivity mediating the association between antenatal mental health problems and behavioral problems in the child (18, 19). A related concept is parental reflective functioning which is defined as the parent's capacity to reflect upon her child's emotional experience and her own experiences as a parent (20). Maternal reflective functioning capacities has been linked to caregiving behavior (e.g. sensitivity) and studies have demonstrated that higher levels of reflective functioning are associated with secure child attachment, better emotion regulation abilities, and less psychopathological symptoms (21).”

2. One of the inclusion criteria is that mothers had to present “psychosocial or social vulnerabilities” identified by the general practitioner. This is too vague and undefined. Please specify in detail the inclusion criteria, since this is a RCT and inclusion criteria are crucial to select the right population to be treated.

The inclusion criteria were made based on the official Danish Health Care recommendations. This was both a clinical and practical decision. Clinically, it was considered an advantage that potential participants were already identified and assessed (based on clinical records, self-reported information from pregnant women, and clinical presentation) by clinical staff as “being in risk” and in need of additional interventions. Practically, this group was referred to a special unit at the hospital where a smaller group of midwifes could be trained to recruit potential participants.

We have made the Participant section more clear with a headline and included the following paragraph in the manuscript (lines 133-136): 

“This is in line with official Danish Health Care recommendations (34). The group consists of women with complex psychological/psychiatric problems and/or difficult social problems with a need for interdisciplinary perinatal care.”

3. The experimental intervention is poorly described. Authors should explain in details the experimental intervention (i.e., how many sessions were provided to mothers, what was the content of sessions, how session were structured, etc.), and report the appropriate citation to the available manuals.

We inserted a reference to the manual and a table (Table 1; line 185) describing number, content and time point of sessions. The reference can found at line 179 in the manuscript.

4. The main outcome was assessed through the CIB. Is this global rating scheme validated? Were raters trained in the use of the assessment instruments? Moreover, authors should specify whether mental health professionals who performed the assessments were blinded to mothers’ allocation to the experimental vs. control group.

More information on the CIB has been added to the Method section (lines 194-201) :

“The CIB has been validated in both longitudinal and intervention studies showing predictive validity for child development outcomes (37, 38), discriminate validity between mothers with and without mental health problems (17), as well as ability to detect effects of perinatal interventions (39, 40). The current study used the maternal sensitivity composite. At approximately nine months, mother-infant interactions were videotaped during a home visit by a researcher not blinded to mothers’ allocation to the experimental vs. control group. A coder trained in the CIB by Ruth Feldman blinded to treatment allocation coded all mother-infant interactions.”

5. I was not able to find thought the manuscript when women were assessed for the first time (I suppose during pregnancy) and when they were reassessed. Could the author better specify this issue?

Information on the procedure and data collection has been added to the manuscript (lines 146-151):

“A researcher contacted eligible participants that consented to be contacted by phone, followed by a home visit to provide further information on the project, obtain informed consent, and enroll interested women and their families in the project. Data collection took place at three time points: at study inclusion at 3–5 months of pregnancy, when the infant is 8 weeks old, and after the intervention has ended when the child is 9 months old. Only data from the baseline assessment and after the intervention are reported in this study.”

6. In the result section, it is reported that 35 out of 76 women presented mental health problems at pregnancy and 28 of them were on psychiatric or psychological treatment. This is a bit confusing, since authors have excluded from the recruitment women with major psychiatric problems. Could you clarify this issue?

As specified in the Participant section, inclusion criteria were pregnant women experiencing complex psychological difficulties and/or psychiatric disorders, such as mild to moderate depression, anxiety, personality disorders and/or ADHD. Exclusion criteria were specific psychiatric disorders (active eating disorder, severe depression, psychosis, schizophrenia, bipolar disorder, and severe obsessive-compulsive disorder) as women with these disorders already received treatment elsewhere. We did not expect women with such severe disorders to be able to participate in two interventions due to limited mental and practical resources.

---

## [Decision Letter · Decision Letter 2]

26 Oct 2022

Efficacy of a Randomized Controlled Trial of a Perinatal Adaptation of COS-P in Promoting Maternal Sensitivity and Mental Wellbeing among Women with Psychosocial Vulnerabilities

PONE-D-21-31294R2

Dear Dr. Røhder,

We’re pleased to inform you that your manuscript has been judged scientifically suitable for publication and will be formally accepted for publication once it meets all outstanding technical requirements.

Kind regards,

Maiken Pontoppidan

Academic Editor

PLOS ONE

Additional Editor Comments (optional):

Thank you for addressing the issues raised by the reviewers.

I have accepted the paper but I suggest that you look at comment 2 and 3 from reviewer 2 and see if you can incorparate these two comments in the final paper.

Reviewers' comments:

Reviewer's Responses to Questions

**Comments to the Author**

1. If the authors have adequately addressed your comments raised in a previous round of review and you feel that this manuscript is now acceptable for publication, you may indicate that here to bypass the “Comments to the Author” section, enter your conflict of interest statement in the “Confidential to Editor” section, and submit your "Accept" recommendation.

Reviewer #1: All comments have been addressed

Reviewer #2: All comments have been addressed

2. Is the manuscript technically sound, and do the data support the conclusions?

Reviewer #1: (No Response)

Reviewer #2: Partly

3. Has the statistical analysis been performed appropriately and rigorously? 

Reviewer #1: (No Response)

Reviewer #2: Yes

4. Have the authors made all data underlying the findings in their manuscript fully available?

Reviewer #1: (No Response)

Reviewer #2: Yes

5. Is the manuscript presented in an intelligible fashion and written in standard English?

Reviewer #1: (No Response)

Reviewer #2: Yes

6. Review Comments to the Author

Reviewer #1: (No Response)

Reviewer #2: Thank you for this revised version of the paper, which is improved compared to the previous one. However, I have further minor comments that should be addressed:

1. in the inclusion criteria, authors state that patients were recruited whether " they were: 1) identified as having psychological or social vulnerabilities by the general practitioner or the hospital. This is in line with official Danish Health Care recommendations. The group consists of women with complex psychological/psychiatric problems and/or difficult social problems with a need for interdisciplinary perinatal care." Although there is an appropriate reference, it would help the reader to specify the kind of difficulties considered. Have you also carried out some psychiatric full-examination in order to assess the presence of any mental disorders?

2. IN the introduction, authors should consider to quote some recent papers published on perinatal mental health (please see e.g., Howard LM, Khalifeh H. Perinatal mental health: a review of progress and challenges. World Psychiatry. 2020 Oct;19(3):313-327. doi: 10.1002/wps.20769. PMID: 32931106; PMCID: PMC7491613; Glover V. Prenatal mental health and the effects of stress on the foetus and the child. Should psychiatrists look beyond mental disorders? World Psychiatry. 2020 Oct;19(3):331-332. doi: 10.1002/wps.20777. PMID: 32931095; PMCID: PMC7491637)

3. in the discussion, authors should add a comment on the feasibility in clinical routine practice of the proposed intervention, also comparing it with other psychosocial interventions available.

7. PLOS authors have the option to publish the peer review history of their article (what does this mean?). If published, this will include your full peer review and any attached files.

Reviewer #1: No

Reviewer #2: No
